# Conditional Moment Alignment for Improved Generalization in Federated Learning

**Jayanth Reddy Regatti**[1]    **Songtao Lu**[2]    **Abhishek Gupta**[1]    **Ness Shroff**[1]

[1]The Ohio State University, [2]IBM Thomas J. Watson Research Center

{regatti.1, gupta.706, shroff.11}@osu.edu
songtao@ibm.com

## Abstract

In this work, we study model heterogeneous federated learning (FL) for classification problems where distributed clients have different model architectures. Unlike existing works focusing on model heterogeneity, we neither require access to a public dataset nor do we impose constraints on the model architecture of clients and ensure that the clients' models and data are private. We show a generalization result, which provides fundamental insights into the role of the representations in FL. Further, we propose a theoretically grounded algorithm, named **Fed**erated **C**onditional **M**oment **A**lignment (Fed-CMA), which is able to align class conditional distributions of each client in the feature space with provable convergence guarantees. Through multiple numerical experiments, we show that Fed-CMA outperforms other baselines on CIFAR-10, MNIST, EMNIST, FEMNIST in the considered setting.

## 1 Introduction

In the paradigm of federated learning (FL), it is quite often that the client models have different architectures due to heterogeneity of computational hardware devices (such as GPU memory, smartphones). In such cases, naive parameter aggregating as in federated averaging (FedAvg) McMahan et al. (2017) or FedProx Li et al. (2020) might be no longer possible for achieving satisfactory generalization performance as in the homogeneous case. Even though some knowledge distillation approaches use a public dataset to address this issue, this type of datasets may not always be available and the role of the public dataset on the generalization performance is not yet understood well. Also, which type of architectures used in deep learning has a huge impact on the performance. The clients may consider the architecture as a trade secret (often due to the amount of resources spent in designing it) and may not be willing to share the model architecture because of the intellectual property concerns. We term this as *model privacy*. This also implies that the users are free to choose any type of (agnostic) model architecture without sharing it. In this work, we consider the data and model heterogeneous FL scenario with facing the following possible additional constraints: (i) **model privacy**, (ii) **no public dataset** and (iii) **unrestricted model space**. See Table 1 for a summary of relevant works. Note that, while model heterogeneity refers to a more general case where the users can have different model spaces such as random forests, decision trees and neural networks, we restrict our study to the use of neural networks (since the only way to learn in the general case is with distillation techniques which require a public dataset). Below, we provide a motivating example for this work.

(Motivating use case) Consider the following application where a product manufacturer works with multiple original equipment manufacturers (OEMs) from different countries (due to regulatory restrictions) for developing a new product line. The product manufacturer intends to improve the diagnostics and prognostics which is often a supervised classification task. Since it is a new product line, data is scarce at every OEM (no public dataset), thus requiring the OEMs to collaborate. The

OEMs may differ in the type of sensors leading to *data heterogeneity* and cannot share the data due to the *data privacy* constraint. They may also have the *model heterogeneity* issue due to various distributed computing resources that restrict the type of model architectures they would deploy. Moreover, in such scenarios, the OEMs should be able to choose the model architecture of their choice and may not be willing to share the architecture.

| Work | MH | MP | PD | RM | C |
|------|----|----|----|----|---|
| Zhang et al. (2021) | ✓ | ✓ | ✓ | ✗ | ✗ |
| Lin et al. (2020) Cho et al. (2022) | ✓ | ✗ | ✓ | ✗ | ✗ |
| Diao et al. (2020) | ✓ | ✗ | ✗ | ✓ | ✗ |
| Yao et al. (2021) Li et al. (2021) | ✓ | ✗ | ✗ | ✓ | ✓ |
| Litany et al. (2022) | ✓ | ✓ | ✗ | ✓ | ✗ |
| Zhu et al. (2021) | ✓ | ✓ | ✗ | ✗ | ✗ |
| **Ours** | ✓ | ✓ | ✗ | ✗ | ✓ |

**Table 1:** MH: Model Heterogeneity, MP: Model Privacy, PD: Public Datset, RM: Restricted Model space, C: Convergence

In this paper, we propose a fundamentally different approach than the works in Table 1 to address model heterogeneity with a new algorithm that is motivated from theoretical results in the *domain adaptation* (DA) literature. In our case since the client models are heterogeneous, we separate the client model architecture as a feature extractor that projects the input data into a latent space (common for all clients) and a classifier that acts on the latent space. We summarize our contributions as follows:

- A thorough theoretical generalization analysis is provided for model homogeneous and model heterogeneous FL with highlighting the difficulty in the latter case. To the best of our knowledge, this is the first theoretical generalization error bound for the model heterogeneous FL setup, which justifies *the advantage of participating in FL* and *the importance of aligning the latent space distributions across the clients*.

- The proposed simple algorithm (Fed-CMA) can align the latent space conditional distributions and the classification weights across all clients in a federated way with convergence guarantees for finding the first-order stationary points to general non-convex problems.

- Multiple detailed numerical experiments are performed under different FL settings over both synthetic and real datasets. It can be observed that Fed-CMA outperforms the considered baselines and achieves reduced communication complexity as compared to other model heterogeneous FL algorithms.

## 2 Preliminaries

### 2.1 Model Heterogeneous Federated Learning

Consider a $K$-class classification task, $\mathcal{T} := \langle \mathcal{D}, c \rangle$, where $\mathcal{D}$ is the distribution on $\mathcal{X} \subset \mathbb{R}^d$ and $c$ is the ground truth labeling function that maps $\mathcal{X}$ to $\mathcal{Y} := \{1, \cdots, K\}$. We consider the model heterogeneous FL problem with $M$ workers where each worker has a individual task $\mathcal{T}_i := \langle \mathcal{D}_i, c \rangle \ \forall i \in [M]$. Each client owns $D_i := \{x_j, y_j\}_j^{N_i}$, such that $x_j \sim \mathcal{D}_i$, $y_j = c(x_j)$. Let $p_i \in [0, 1]$ such that, $N_i = p_i N$, where $N = \sum_{i=1}^M N_i$. Also, each client has a model $\mathbf{w}_i$ in a hypothesis class $\mathcal{W}_i \subset \mathbb{R}^{d_i}$ (where $d_i$ is the dimensionality of the model space $\mathcal{W}_i$) and a function $f_i : \mathcal{W}_i \times \mathcal{X} \to \mathcal{Y}$ which is used to make a prediction on a given data point. We denote $\mathbf{w}_{(i,t)}$ as the weights at client $i$ at time $t$. We simply write $\mathbf{w}_i$ when time is clear from context and $\mathbf{w}$ when worker and time are clear from context. Given a loss function $l : \mathcal{Y} \times \mathcal{Y} \to \mathbb{R}$, we define the loss function $l_i : \mathcal{W}_i \times \mathcal{X} \times \mathcal{Y} \to \mathbb{R}$ for each worker as $l_i(\mathbf{w}_i, x, y) = l\big(f_i(\mathbf{w}_i, x), y\big)$ . Given

a finite dataset $D_i = \{(x_j, y_j)\}_{j=1}^{N_i}$, we define the population loss (true loss) $L_i$ and the empirical loss $\widehat{L}_i$ (the subscript $i$ refers to $\mathcal{D}_i$ at the worker $i$) as $L_i(\mathbf{w}_i) = \mathbb{E}_{(x,y)\sim\mathcal{D}_i}[l_i(\mathbf{w}_i, x, y)]$ and $\widehat{L}_i(\mathbf{w}, D_i) = \frac{1}{N_i} \sum_{j=1}^{N_i} l_i(\mathbf{w}, x_j, y_j)$. We simply use $\widehat{L}_i(\mathbf{w})$ when $D_i$ is clear from context. We use the terms "loss", error and risk interchangeably. The objective of model heterogeneous FL is to simultaneously learn $\{\mathbf{w}_i^*\}_{i=1}^M$ such that

$$(\mathbf{w}_1^*, \mathbf{w}_2^*, \cdots, \mathbf{w}_M^*) = \arg\min_{\{\mathbf{w}_i\}_{i=1}^M} \sum_{i=1}^M \alpha_i L_i(\mathbf{w}_i), \tag{1}$$

where $\alpha_i \geq 0$, $\sum_{i=1}^M \alpha_i = 1$. Popular choices for $\alpha_i$ are $\frac{1}{M}$ or $p_i$.

The challenge in solving (1) is because $\{\mathbf{w}_i\}_{i=1}^M$ share different network architectures, meaning that the existing algorithms on parameter aggregation (such as FedAvg, FedProx) cannot be used. Under model heterogeneity, we only assume a common latent space $(\mathcal{Z} \subseteq \mathbb{R}^{d_e})$ for all workers. In other words, each worker $i$ has a model $\mathbf{w}_i = (\mathbf{u}_i, \mathbf{v}_i) \in \mathcal{W}_i$ where $\mathbf{u}_i \in \mathcal{U}_i$, $\mathbf{v}_i \in \mathcal{V}$. We define the prediction function as $f_i(\mathbf{w}_i, x) := h(\mathbf{v}_i, g_i(\mathbf{u}_i, x))$ where $g_i(\mathbf{u}_i, \cdot) : \mathcal{X} \to \mathcal{Z}$ projects $\mathcal{X}$ to the latent space and $h(\mathbf{v}_i, \cdot) : \mathcal{Z} \to \mathcal{Y}$ makes the prediction. In general, $\dim(\mathcal{U}_i) \gg \dim(\mathcal{V})$. Typically, consider a neural network where $\mathbf{u}_i$ corresponds to the weights of the feature extractor and $\mathbf{v}_i$ corresponds to the weights of the classification layer (see Fig 1). For example, Pillutla et al. (2022) and Liang et al. (2020) consider a similar setup for personalizing the input representation learning layers while sharing the classification layer with the server.

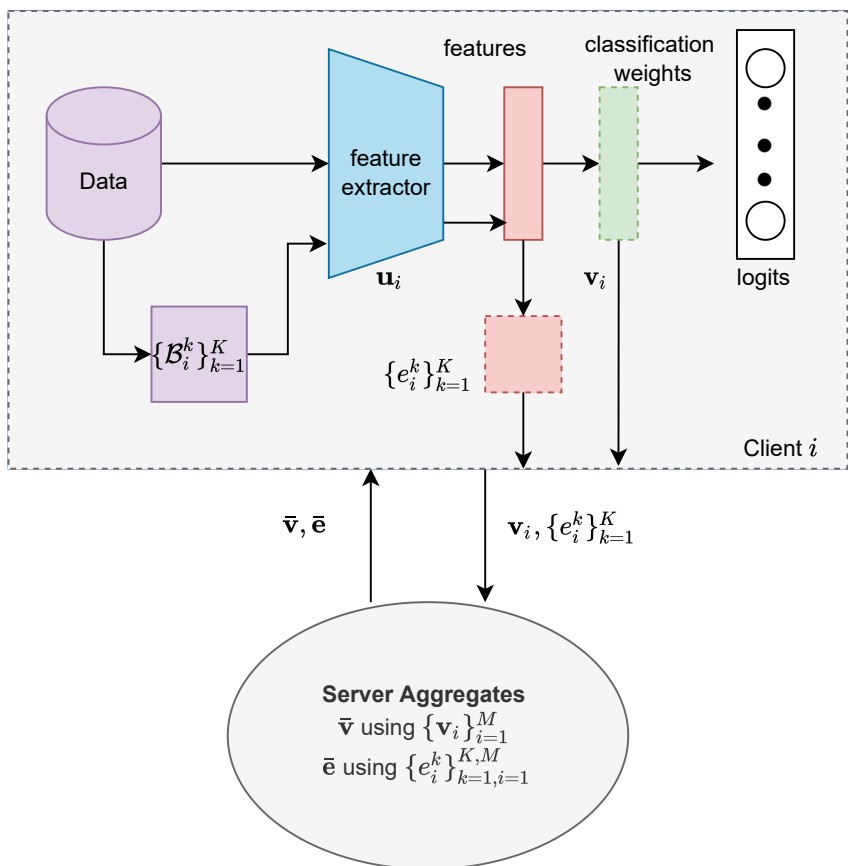

**Figure 1:** Illustration of Fed-CMA .

## 2.2 Domain Adaptation

In DA, a classifier is trained on a sufficiently large source dataset (task) such that it is expected to perform well on a target dataset (task) with few or no labeled data points. For the purpose

of understanding the DA perspective and studying the generalization, we shall consider a binary classification task. Let $\mathcal{X} \subset \mathbb{R}^d$ and $\mathcal{Y} = [0,1]$, which is construed as the probability of having the label 0. Let us denote the random variables in $\mathcal{X}, \mathcal{Y}$ as $X$ and $Y$ respectively. We use the terms "task", "domain" and "client" interchangeably from hereon. Consider that the clients have the same model class (or hypothesis) denoted by $\mathcal{W}$ and hence the same prediction function $f : \mathcal{W} \times \mathcal{X} \to \mathcal{Y}$. We define the disagreement between two models $\mathbf{w}, \mathbf{w}' \in \mathcal{W} \subseteq \{0,1\}^{\mathcal{X}}$ as $L_i(\mathbf{w}, \mathbf{w}') = \mathbb{E}_{x \in \mathcal{D}_i}\big[|f(\mathbf{w}, x) - f(\mathbf{w}', x)|\big]$. Suppose that the target labeling functions $\{c_i : \mathcal{X} \to \mathcal{Y}\}$ are different for every client, then the error of a model $\mathbf{w} \in \mathcal{W}$ on a client $i$ is $L_i(\mathbf{w}) := L_i(\mathbf{w}, c_i)$. The error of a model $\mathbf{w}$ on clients $i, j$ can be related as Zhao et al. (2019)

$$L_j(\mathbf{w}) \leq L_i(\mathbf{w}) + d_{\mathcal{W}\Delta\mathcal{W}}(\mathcal{D}_i, \mathcal{D}_j) + \min\Big\{\mathbb{E}_{x \in \mathcal{D}_i}\big[|c_i(x) - c_j(x)|\big], \mathbb{E}_{x \in \mathcal{D}_j}\big[|c_i(x) - c_j(x)|\big]\Big\}, \tag{2}$$

where $d_{\mathcal{W}\Delta\mathcal{W}}$ is used to measure the divergence between the two distributions $\mathcal{D}_i, \mathcal{D}_j$ Kifer et al. (2004) (see the appendix for formal definitions). If the divergence is small, it is hard to discriminate between the two data distributions. This shows that, if the labeling functions $c_i, c_j$ are close, and $d_{\mathcal{W}\Delta\mathcal{W}}$ is small, then a model trained on one client $i$ (source) can perform well on another client $j$ (target). Let $q_i(X, Y)$ denote the joint distribution of $X, Y$ for client $i$. Suppose that the labeling functions $\{c_i\}$ are the same for all the clients, then $q_i(Y|X) = q_j(Y|X)$, but the marginals of $X$ are not equal. This is called the *covariate shift* assumption Johansson et al. (2019) which we consider in this work. Under this assumption, the last term in (2) is zero and only $d_{\mathcal{W}\Delta\mathcal{W}}$ needs to be small. However, given two tasks, the divergence between the distributions in $\mathcal{X}$ is fixed. This motivated the learning of *domain invariant representations* such that the divergence of distributions in the representation space (or latent space) is minimized. Broadly, these methods take the following approaches. **(i) Minimizing an integral probability metric (IPM):** Along with the classification loss on source data, the network is explicitly trained to minimize an IPM such as maximum mean discrepancy (MMD Gretton et al. (2006)) Long et al. (2013); Tzeng et al. (2014); Long et al. (2015) or a central moment discrepancy (CMD) Zellinger et al. (2017) between the source and target datasets at the representation layer. **(ii) Adversarial methods:** Ganin et al. (2016); Bousmalis et al. (2016); Tzeng et al. (2017) train the network by minimizing the label classification loss and maximizing a domain classification loss.

Let $\mathcal{Z} \subset \mathbb{R}^{d_e}$ be a latent space, $g(\mathbf{u}, \cdot) : \mathcal{X} \to \mathcal{Z}$ be a representation function where $\mathbf{u} \in \mathcal{U}$ and $Z$ be a random variable in $\mathcal{Z}$. Given a task $\langle \mathcal{D}, c \rangle$, $\mathbf{u}$ induces the distribution $\tilde{\mathcal{D}}(\mathbf{u})$ on $\mathcal{Z}$ and the labeling function $\tilde{c}_{\mathbf{u}} : \mathcal{Z} \to \mathcal{Y}$ such that $\mathbb{E}_{z \in \tilde{\mathcal{D}}(\mathbf{u})}\big[\mathbb{I}_B(z)\big] = \mathbb{E}_{x \in \mathcal{D}}\big[\mathbb{I}_B(g(\mathbf{u}, x))\big]$, where $B \subset \mathcal{Z}$ is a measurable set, $\mathbb{I}_B$ is an indicator function over $B$, and $\tilde{c}_{\mathbf{u}}(z) := \mathbb{E}_{x \in \mathcal{D}}\big[c(x)|g(\mathbf{u},x) = z\big], \forall z \in \mathcal{Z}$. Let $q_i(Z, Y)$ be the joint distribution at client $i$ using a representation $\mathbf{u}_i$. Under this setup, the induced labeling functions in the latent space may not be equal even when the labeling functions in the input space are the same Zhao et al. (2019) and several works followed to address this shortcoming by assuming a *generalized label shift* condition Tachet des Combes et al. (2020); Shui et al. (2021) where $q_i(Z|Y) = q_j(Z|Y)$. This motivates our conditional alignment technique to solve (1). In contrast with DA (even the multi-source) setting, we have the following challenges: **(i) restriction on data sharing and model sharing, (ii) learning simultaneously for all clients**. In classic multi-source DA, the goal is to train a classifier on all sources so that it can generalize well on the target task. In our Federated setting, every client simultaneously solves a multi-source DA problem viewing itself as the target. On the other hand, each client has a labelled dataset unlike in unsupervised DA, where the target has no labeled data and one might need to estimate label distribution ratios.

## 3 Generalization Result for Model Homogeneous and Heterogeneous FL

Next, we will first discuss the existing results for generalization in FL and highlight the limitations of these results. Let us consider the case where all the client models are homogeneous, and the client weights can be shared. The weighted data distribution is denoted as $\mathcal{D}_{\boldsymbol{\alpha}} = \sum_{i=1}^{M} \alpha_i \mathcal{D}_i$. Let $\boldsymbol{\alpha} = (\alpha_1, \alpha_2, \cdots, \alpha_M)$ be the domain weights for each user such that $\sum_{i=1}^{M} \alpha_i = 1$, and let $j$ be the target domain (which can be one of the clients). There are two types of existing generalization results for FL in the literature where the final prediction (i) uses a weighted average of the client model weights i.e., $f\big(\sum_i \alpha_i \mathbf{w}_i, x\big)$ (Peng et al., 2019), (ii) uses an ensemble of the predictions on the client models, i.e., $\sum_i \alpha_i f_i(\mathbf{w}_i, x)$ (Lin et al., 2020; Cho et al., 2022).

**Averaged model weights** Following Section 2.1, the empirical loss is defined as $\widehat{L}_i(\mathbf{w}_i) := \frac{1}{N_i} \sum_{(x,y) \in D_i} |f(\mathbf{w}_i, x) - y|$ and the $\boldsymbol{\alpha}$ empirical weighted loss as $\widehat{L}_{\boldsymbol{\alpha}}(\mathbf{w}) = \sum_{i=1}^M \alpha_i \widehat{L}_i(\mathbf{w})$. The true weighted loss $L_{\boldsymbol{\alpha}}$ can be defined similarly by following Section 2.1. Theorem 2 of Peng et al. (2019) shows a generalization bound for FL (when $j$ is the target): with probability at least $1 - \delta$,

$$L_j(\mathbf{w}_j) \leq \widehat{L}_{\boldsymbol{\alpha}}\Big(\sum_{i=1}^M \alpha_i \mathbf{w}_i\Big) + \sum_{i=1}^M \alpha_i\Big(\frac{1}{2} d_{\mathcal{W}\Delta\mathcal{W}}(\mathcal{D}_i, \mathcal{D}_j)\Big) + \mathcal{O}\Big(\frac{\log\frac{1}{\delta}}{\sqrt{N}}\Big). \qquad (3)$$

Here, $\mathcal{O}(\cdot)$ hides logarithmic terms and model class complexities that are constant. For the same model class, if the target domain $j$ only uses the data available locally, from standard results in generalization, we know that the user's generalization error can be bounded as $L_j(\mathbf{w}_j) \leq \widehat{L}_j(\mathbf{w}_j) + \mathcal{O}\Big(\frac{\log 1/\delta}{\sqrt{N_j}}\Big)$ with probability at least $1 - \delta$. The benefit of joining FL can be see from the $\frac{1}{\sqrt{N}}$ term in (3) since $N \gg N_j$ when $M$ is large. The $\widehat{L}_{\boldsymbol{\alpha}}(\sum_i \alpha_i \mathbf{w}_i)$ (empirical loss at all users) and $d_{\mathcal{W}\Delta\mathcal{W}}$ (captures heterogeneity among users) terms must be smaller than $\widehat{L}_j(\mathbf{w}_j)$ for this benefit to be realizable. This encourages the FL objective to solve $\arg\min \widehat{L}_{\boldsymbol{\alpha}}(\mathbf{w})$ by sharing the model weights.

However, this result has a major drawback. It only holds for model homogeneous FL, since the result is with respect to the weighted model $\sum_{i=1} \alpha_i \mathbf{w}_i$, which is clearly not possible in model heterogeneous (or model private) FL.

**Ensembled model predictions** We slightly abuse the notation for this discussion by defining the loss as follows: $\widehat{L}_i(f_i) := \frac{1}{N_i} \sum_{(x,y) \in D_i} |f_i(\mathbf{w}_i, x) - y|$. The generalization result where the target distribution is $\mathcal{D}_j$ can be summarized as follows (although there are subtle variations in Theorem 5.1 in Lin et al. (2020) and Theorem 1 in Cho et al. (2021)): with probability at least $1 - \delta$,

$$L_j\Big(\sum_{i=1}^M \alpha_i f_i\Big) \leq \sum_{i=1}^M \alpha_i \widehat{L}_i(f_i) + \sum_{i=1}^M \frac{\alpha_i}{2} d_{\mathcal{W}\Delta\mathcal{W}}(\mathcal{D}_i, \mathcal{D}_j) + \sum_{i=1}^M \mathcal{O}\Big(\frac{\alpha_i \log\frac{1}{\delta}}{\sqrt{N_i}}\Big). \qquad (4)$$

Although this result holds for the model heterogeneous case, the result fails to capture the advantage of performing FL for the following reasons. Firstly, the bound contains the term $\sum_{i=1}^M \mathcal{O}\Big(\frac{1}{\sqrt{N_i}}\Big)$ which is worse than $\mathcal{O}\Big(\frac{1}{\sqrt{N_j}}\Big)$, while also containing the divergence terms $d_{\mathcal{W}\Delta\mathcal{W}}$ and the local loss terms $\sum_{i \neq j} \alpha_i \widehat{L}_i(f_i)$. Therefore, the result does not show any improvement over local training. The result also holds when the models are trained locally and the ensemble is used only during prediction. This does not motivate the distillation loss (with communication during training) in Lin et al. (2020); Cho et al. (2022). Even more importantly, the impact of the public dataset distribution (used for distillation) is not captured in the result. Therefore, this bound is uninformative and fails to justify the benefit of FL.

Also note that, the original results in Peng et al. (2019); Lin et al. (2020); Cho et al. (2022) include an additional term $\lambda$ which quantifies the best oracle loss. However, we observe that this $\lambda$ term can be avoided and thus the we omitted it from the generalization results in (3) and (4).

We shall now address the limitations mentioned above by (i) decoupling the representation layer from the classification layer which allows us to study model heterogeneity in FL and (ii) using results from multi-source domain adaptation which allows us to show $\frac{1}{\sqrt{N}}$ dependence without assuming a centralized dataset. While Theorem 1 of Zhu et al. (2021) follows (i), they still show a $\frac{1}{\sqrt{N_i}}$ dependence and only consider the model homogeneous case.

### 3.1 Model Homogeneity

Let us first show the result for the model homogeneous case, where averaging the model weights is optimal. Given a representation function $\mathbf{u}$, denote the induced distribution of the $i^{th}$ client as $\tilde{\mathcal{D}}_i(\mathbf{u})$ and the induced labeling function as $\tilde{c}_{\mathbf{u}}$. The empirical risk of the $i^{th}$ client for the model $(\mathbf{u}, \mathbf{v})$ is defined as $\widehat{L}_{i,\mathbf{u}}(\mathbf{v}, \tilde{c}_{\mathbf{u}}) := \frac{1}{N_i} \sum_{x \in D_i} |h(\mathbf{v}, g(\mathbf{u}, x)) - \tilde{c}_{\mathbf{u}}(g(\mathbf{u}, x))|$ and the true risk $L_{i,\mathbf{u}}(\mathbf{v}, \tilde{c}_{\mathbf{u}})$

is the expectation of $\widehat{L}_{i,\mathbf{u}}(\mathbf{v}, \tilde{c}_{\mathbf{u}})$ taken with respect to the draw of $D_i$. The domain weighted empirical risk is defined similarly as $\widehat{L}_{\boldsymbol{\alpha}}(\mathbf{v}) = \sum_{i=1}^{M} \alpha_i \widehat{L}_{i,\mathbf{u}}(\mathbf{v}, \tilde{c}_{\mathbf{u}})$ and the true weighted risk is defined similarly as $L_{\boldsymbol{\alpha}}(\mathbf{v})$. We denote the mixture of the induced client distributions defined as $\tilde{\mathcal{D}}_{\boldsymbol{\alpha}} = \sum_{i=1}^{M} \alpha_i \tilde{\mathcal{D}}_i(\mathbf{u})$. Let the target distribution be given by $\mathcal{D}_j$ and the induced distribution is $\tilde{\mathcal{D}}_j(\mathbf{u})$. We now state the generalization bound.

**Theorem 1** (Model Homogeneity). *Given* $\mathbf{u}$, *let* $\widehat{\mathbf{v}} = \arg\min_{\mathbf{v} \in \mathcal{V}} \widehat{L}_{\boldsymbol{\alpha}}(\mathbf{v})$ *and* $\mathbf{v}_j^* = \arg\min_{\mathbf{v} \in \mathcal{V}} L_{j,\mathbf{u}}(\mathbf{v}, \tilde{c}_{\mathbf{u}})$. *Then for any* $\delta > 0$, *w.p.* $> 1 - \delta$, *we have*

$$L_{j,\mathbf{u}}(\widehat{\mathbf{v}}) - L_{j,\mathbf{u}}(\mathbf{v}_j^*) \leq 4\sqrt{\sum_{i=1}^{M} \frac{\alpha_i^2}{p_i}} \mathcal{O}\left(\sqrt{\frac{\log\frac{1}{\delta}}{2N}}\right) + B,$$

*where* $B = \sum_{i=1}^{M} \alpha_i d_{\mathcal{V}\Delta\mathcal{V}}\left(\tilde{\mathcal{D}}_i(\mathbf{u}), \tilde{\mathcal{D}}_j(\mathbf{u})\right)$.

Let $\alpha_i = p_i$, then the generalization error decays as $\mathcal{O}\left(\sqrt{1/N}\right)$. Therefore, the dependence on the number of data points shows that the client attains better generalization error by participating in the FL setup provided the divergence term is small. Note that, as compared to (3), we have divergence between the induced distributions $d_{\mathcal{V}\Delta\mathcal{V}}(\tilde{\mathcal{D}}_i(\mathbf{u}), \tilde{\mathcal{D}}_j(\mathbf{u}))$ as compared to $d_{\mathcal{W}\Delta\mathcal{W}}(\mathcal{D}_i, \mathcal{D}_j)$ which is fixed for the given client datasets. The divergence term in Theorem 1 can be minimized by learning invariant feature representations $\mathbf{u}$ which is the topic of interest in domain invariant representation learning Long et al. (2015); Zhao et al. (2019).

## 3.2 Model Heterogeneity

For the model heterogeneous case (since there is no global model), let $\mathbf{u}_i$ be the representation layer weights of the $i$th user. We then define the induced distributions $\tilde{\mathcal{D}}_i(\mathbf{u}_i)$, labeling functions $\tilde{c}_{\mathbf{u}_i}$, empirical risk $\widehat{L}_{i,\mathbf{u}_i}(\mathbf{v}_i, \tilde{c}_{\mathbf{u}_i})$ with respect to $\mathbf{u}_i$ for every client. The weighted empirical risk is then $\widehat{L}_{\boldsymbol{\alpha}}(\mathbf{v}) = \sum_{i=1}^{M} \alpha_i \widehat{L}_{i,\mathbf{u}_i}(\mathbf{v}, \tilde{c}_{\mathbf{u}_i})$. A major difference is that, the induced labeling functions $\tilde{c}_{\mathbf{u}_i}$ are different for every client whereas in Theorem 1, it is the same for all clients due to shared $\mathbf{u}$. Let the target distribution be given by $\mathcal{D}_j$ and the corresponding representation be $\mathbf{u}_j$ that induces $\tilde{\mathcal{D}}_j(\mathbf{u}_j)$ and $\tilde{c}_{\mathbf{u}_j}$. We now state the generalization bound as follows.

**Theorem 2** (Model Heterogeneity). *Let* $\widehat{\mathbf{v}} = \arg\min_{\mathbf{v} \in \mathcal{V}} \widehat{L}_{\boldsymbol{\alpha}}(h)$ *and* $\mathbf{v}_j^* = \arg\min_{\mathbf{v} \in \mathcal{V}} L_{j,\mathbf{u}_j}(\mathbf{v}, \tilde{c}_{\mathbf{u}_j})$ *be the minimizer of the true target risk. Then for any* $\delta > 0$, *w.p.* $> 1 - \delta$, *we have*

$$L_{j,\mathbf{u}_j}(\widehat{\mathbf{v}}) - L_{j,\mathbf{u}_j}(\mathbf{v}_j^*) \leq 4\sqrt{\sum_{i=1}^{M} \frac{\alpha_i^2}{p_i}} \mathcal{O}\left(\sqrt{\frac{\log\frac{1}{\delta}}{N}}\right) + 2A + B,$$

*where* $A = \sum_{i=1}^{M} \alpha_i \min\left\{\mathbb{E}_{z \in \tilde{\mathcal{D}}_i(\mathbf{u}_i)}\left[|\tilde{c}_{\mathbf{u}_i}(z) - \tilde{c}_{\mathbf{u}_j}(z)|\right], \mathbb{E}_{z \in \tilde{\mathcal{D}}_j(\mathbf{u}_j)(z)}\left[|\tilde{c}_{\mathbf{u}_i}(z) - \tilde{c}_{\mathbf{u}_j}(z)|\right]\right\}$ *and* $B = \sum_{i=1}^{M} \alpha_i d_{\mathcal{V}\Delta\mathcal{V}}(\tilde{\mathcal{D}}_i(\mathbf{u}_i), \tilde{\mathcal{D}}_j(\mathbf{u}_j))$.

The bound is similar to that of Theorem 1 with two key differences: (i) the addition of term $A$ and (ii) the divergence in $B$ is computed between the induced distributions on private representations $\mathbf{u}_i$ of the clients. These two differences are inherently due to the presence of model heterogeneity. The term $A$ determines the closeness of $\tilde{c}_{\mathbf{u}_j}$ with every other $\tilde{c}_{\mathbf{u}_i}$. Observe that in the model homogeneous case where $\mathbf{u}$ is shared among the clients, $A = 0$ since all the induced labeling functions are equal to $\tilde{c}_{\mathbf{u}}$. Similarly, as compared to the model homogeneous case, $B$ here is measured between the $\tilde{\mathcal{D}}_i(\mathbf{u}_i)$ and $\tilde{\mathcal{D}}_j(\mathbf{u}_j)$ for $\mathbf{u}_i \neq \mathbf{u}_j$ leading to a higher divergence. Due to these differences, achieving good generalization in model heterogeneous FL is more challenging than in model homogeneous FL. Observe that, in Theorem 1 and 2, the error measured is dependent on the representations since $\widehat{\mathbf{v}}, \mathbf{v}_j^*$ are dependent on the representations. Showing the result without fixing the representations is more challenging and we leave this as future work.

While FL research primarily focuses on improving the convergence rate by addressing issues such as variance reduction or client-drift Karimireddy et al. (2020), the role of the feature extractors in reducing the degree of heterogeneity in the latent space receives rare attention. To the best of our knowledge, our work is the first to show generalization bounds for FL, which provide these fundamental insights and highlight the benefits of participating in FL.

---

**Algorithm 1** Fed-CMA

1: **At Server:**
2: **for** $t = 1, \cdots, T$ **do**
3:     Collect mean embeddings $\{e_{(i,t)}^k\}_{k=1}^K$ and classification layer weights $\mathbf{v}_{(i,t)}$ for all $i \in [M]$
4:     Update $\bar{\mathbf{e}}$ and $\bar{\mathbf{v}}$ using (7)
5:     Broadcast $\{e_{(t)}^k\}_{k=1}^K$ and $\bar{\mathbf{v}}_{(t)}$ to the workers.
6: **end for**
7: **At Worker** $i$**:**
8: **for** $t = 1, \cdots, T$ **do**
9:     Receive $\{e_{(t)}^k\}_{k=1}^K$ and $\bar{\mathbf{v}}_{(t)}$ from the server.
10:     Pick random minibatch $\xi_{(t)}$ from $D_i$ and compute $\nabla \widehat{\Phi}_i$.
11:     Update $\mathbf{w}_{(i,t)}$ according to (6)
12:     Compute $\{e_{(i,t)}^k\}_{k=1}^K$ using $e_i^k = \frac{1}{b_i^k} \sum_{x \in \mathcal{B}_i^k} g_i(\mathbf{u}_{(i,t)}, x)$ for $k \in [K]$.
13:     Share with server: mean embeddings $\{e_{(i,t)}^k\}_{k=1}^K$ and classification layer weights $\mathbf{v}_{(i,t)}$.
14: **end for**

---

## 4   Conditional Moment Alignment

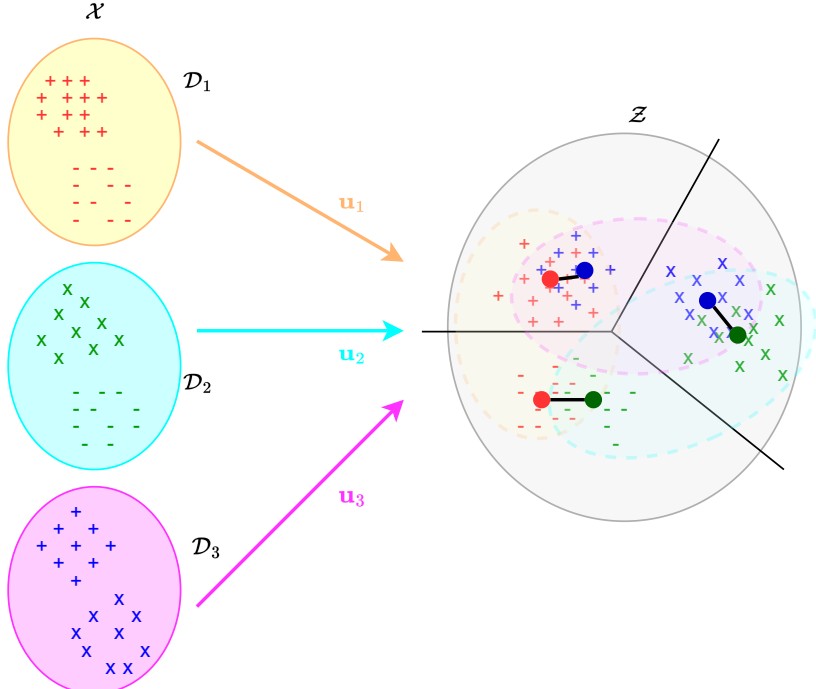

**Figure 2:** Illustration of conditional alignment

We recall the example from Zhao et al. (2019) (Fig 1) which shows that marginal alignment of distributions is insufficient when the labeling functions are different. The goal of representation learning should therefore be to simultaneously reduce the terms $A$ and $B$. Towards this, we propose conditional alignment of distributions in the latent space which aims to learn representation weights $\{\mathbf{u}_i\}$ such that the induced labeling functions are closer and the latent space distributions are closer (see Fig 2 for an illustration).

Although the ideal solution is to align the conditional distributions, it is impractical in the distributed setting. Therefore, we approximate it by aligning only the first order moments of the class conditional mean embeddings of each client and maintain a global variable that tracks the weighted average of the individual mean embeddings. Let $p_i^k$ denote the fraction of data points of label $k$ available at

client $i$ out of all data points of label $k$ at all clients. At the beginning of the training, each client $i$ samples a fixed i.i.d minibatch $\mathcal{B}_i^k \sim \mathcal{D}_i$ for each $k$th label in the dataset $D_i$ with $b_i^k$ points. The size of $b_i^k$ can be much smaller than the size of the original dataset $N_i$. This minibatch is used throughout the training. The class conditional mean embedding vector $e_i^k \in \mathbb{R}^{d^e}$ is computed as $e_i^k = \frac{1}{b_i^k} \sum_{x \in \mathcal{B}_i^k} g_i(\mathbf{u}_i, x)$. We abuse the notation slightly, by using $e_i^k(\mathbf{u})$ when the parameter $\mathbf{u}$ is of interest. These vectors are the class conditional empirical means of each class available at each client.

We maintain a global variable $\mathbf{e} = \{e^k\}_{k=1}^K$ that stores a weighted average of the means of all the clients. This global variable is used to achieve consensus among the clients and align the client distributions in the latent space. Similarly, we maintain a global variable $\bar{\mathbf{v}}$ for the classification layer weights. The population local loss at worker $i$ is

$$\Phi_i(\mathbf{w}_i, \bar{\mathbf{v}}, \mathbf{e}) = L_i(\mathbf{w}_i) + \frac{\lambda_1}{2} p_i \left\| \mathbf{v}_i - \bar{\mathbf{v}} \right\|_2^2 + \frac{\lambda_2}{2} \sum_{k=1}^K \frac{p_i^k}{b_i^k} \sum_{x \in \mathcal{B}_i^k} \left\| g_i(\mathbf{u}_i, x) - e^k \right\|_2^2. \tag{5}$$

Let $\mathbf{w}_{(i,t)} := (\mathbf{u}_{(i,t)}, \mathbf{v}_{(i,t)})$ be the parameters at $i$th worker and $\eta_{(t)}$ be the learning rate at time $t$. We define $\mathbf{e}_t = \{e_{(t)}^k\}_{k=1}^K$ and $\bar{\mathbf{v}}_{(t)}$. Note that we use $(t)$ to highlight that the subscript refers to the time. In Fed-CMA, we perform the following updates on $\mathbf{w}_{(i,t)}, \bar{\mathbf{v}}_{(t)}, \mathbf{e}_{(t)}$ simultaneously,

$$\mathbf{w}_{(i,t+1)} = \mathbf{w}_{(i,t)} - \eta_{(t)} \nabla_{\mathbf{w}_i} \widehat{\Phi}_i(\mathbf{w}_{(i,t)}, \bar{\mathbf{v}}_{(t)}, \mathbf{e}_{(t)}); \tag{6}$$

$$\begin{cases} \bar{\mathbf{v}}_{(t+1)} & = \bar{\mathbf{v}}_{(t)}(1 - \lambda_1 \eta_{(t)}) + \lambda_1 \eta_{(t)} \sum_{i=1}^M p_i \mathbf{v}_{(i,t)}; \\ e_{(t+1)}^k & = e_{(t)}^k(1 - \lambda_2 \eta_{(t)}) + \lambda_2 \eta_{(t)} \sum_{i=1}^M p_i^k e_i^k(\mathbf{u}_{(i,t)}); \end{cases} \tag{7}$$

where the empirical gradient $\widehat{\Phi}_i$ is computed using a minibatch $\xi_i$ that is drawn i.i.d. from $D_i$, $0 < \lambda_1 \eta_{(t)} \leq 1$ and $0 < \lambda_2 \eta_{(t)} \leq 1$. A detailed implementation of Fed-CMA is shown in Algorithm 1.

## 5 Convergence Results

Let $\mathbf{a}$ denote the concatenation of $(\{\mathbf{w}_i\}_{i=1}^M, \bar{\mathbf{v}}, \mathbf{e})$, where all $\{\mathbf{w}_i\}_{i=1}^M, \bar{\mathbf{v}}, \mathbf{e}$ are vectorized so that we have $\mathbf{a} \in \mathbb{R}^{Md_i + 2(d_e+1)K}$. Then, we can write the global objective as

$$\arg \min_{\mathbf{a}} \Phi(\mathbf{a}) := \arg \min_{\{\mathbf{w}_i\}_{i=1}^M, \bar{\mathbf{v}}, \bar{\mathbf{e}}} \sum_{i=1}^M \Phi(\mathbf{w}_i, \bar{\mathbf{v}}, \mathbf{e}). \tag{8}$$

Unless otherwise specified, $\|\cdot\|$ is $\|\cdot\|_2$ for vectors and $\|\cdot\|_F$ for matrices. Next, we make the following assumptions on the loss functions and embedding maps.

**Assumption 1.** *The local loss functions $\{L_i(\mathbf{w}_i)\}_{i=1}^N$ are $\beta_l$ Lipschitz continuous, and $\beta_s$ smooth in $\mathbf{w}_i$. Note that $\|\mathbf{w}_i\|^2 = \|\mathbf{u}_i, \mathbf{v}_i\|^2 = \|\mathbf{u}_i\| + \|\mathbf{v}_i\|$. The loss functions $L_i$ are lower bounded uniformly by a scalar $L_{\inf}$.*

**Assumption 2.** *The function $g_i(\mathbf{u}_i, x)$ is $\beta_e$ Lipschitz continuous and $\beta_g$ smooth $\forall i \in [M]$ with respect to $\mathbf{u}_i$. Moreover, we assume that $g_i(\mathbf{u}_i, x) \in [0,1]^{d_e}, \forall i$.*

**Assumption 3.** *The empirical gradient is an unbiased gradient of the population loss i.e., $\mathbb{E}[\nabla_{\mathbf{a}} \widehat{\Phi}(\{\mathbf{w}_i\}_{i=1}^M, \bar{\mathbf{v}}, \mathbf{e})] = \nabla_{\mathbf{a}} \Phi(\{\mathbf{w}_i\}_{i=1}^M, \bar{\mathbf{v}}, \mathbf{e})$. The variance of the stochastic gradient is bounded, i.e., $\mathbb{E}\left[\left\| \nabla_{\mathbf{a}} \widehat{\Phi}(\{\mathbf{w}_i\}_{i=1}^M, \bar{\mathbf{v}}, \mathbf{e}) - \nabla_{\mathbf{a}} \Phi(\{\mathbf{w}_i\}_{i=1}^M, \bar{\mathbf{v}}, \mathbf{e}) \right\|^2\right] \leq G_1 \left\| \nabla_{\mathbf{a}} \Phi(\{\mathbf{w}_i\}_{i=1}^M, \bar{\mathbf{v}}, \mathbf{e}) \right\|^2 + G_2^2.$*

Assumption 1 is commonly used in analyzing the convergence behavior of stochastic gradient descent (SGD) for general non-convex problems. Assumption 2 implies that the output of $g_i(\mathbf{u}_i, x)$ is bounded and this is easily satisfied in deep neural networks where the activation function is a sigmoid function. While the boundedness of the activation function is not necessary, it makes some parts of the proof simpler. The more general ReLU activation can be used by making some changes to the loss function (more details are provided in the supplementary material). Assumption 3 follows from assuming that the local dataset at each client $i$ is drawn i.i.d from $\mathcal{D}_i$. For simplicity, we consider only one local update (6) before computing the global update (7). We now state the convergence theorem.

**Theorem 3.** *Let Assumptions 1, 2, and 3 hold and run the algorithm with $T$ timesteps. If we chose a constant learning rate $\eta = \sqrt{\frac{2C_1}{TC_0 G_2^2}}$ that satisfies $0 < \eta \leq \min\left\{\frac{1}{\lambda_1}, \frac{1}{\lambda_2}, \frac{1}{C_0(G_1+1)}\right\}$, where $\mathbb{E}[\Phi(\mathbf{a}_{(1)})] - \mathbb{E}[\Phi(\mathbf{a}_{(T)})] \leq \mathbb{E}[\Phi(\mathbf{a}_{(1)})] - \Phi^* \leq C_1$, and $C_0$ is a problem specific constant, then we have $\min_{t:1,\cdots,T} \mathbb{E}\left[\|\nabla_\mathbf{a}\Phi(\mathbf{a}_{(t)})\|^2\right] \leq \sqrt{\frac{2C_1 C_0 G_2^2}{T}}$.*

The proof of the result is provided in the supplementary material. The result shows that, while following the updates in (6) and (7) to solve (8), Fed-CMA can find the first-order stationary point in $\mathcal{O}(1/\sqrt{T})$ updates (which matches the convergence rate of traditional SGD in FL for solving non-convex objectives). Note that, among the model heterogeneous FL works in Table 1, only Yao et al. (2021); Li et al. (2021) show the convergence of their proposed algorithms.

## 6 Numerical Simulations

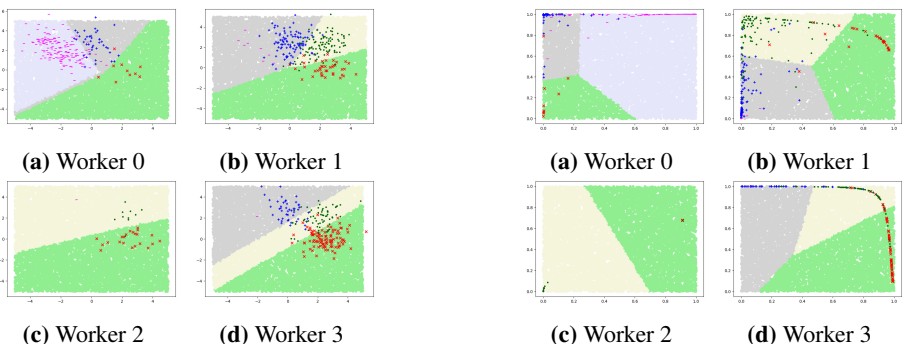

**Figure 4:** Local input space decision boundaries     **Figure 5:** Local latent space decision boundaries

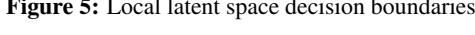

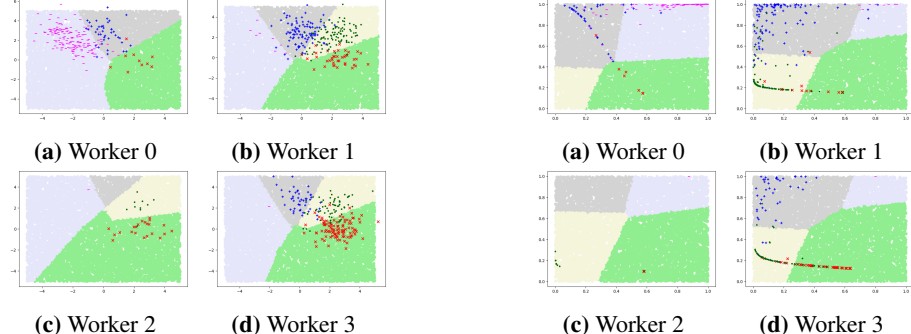

**Figure 7:** FedCMA input space decision boundaries     **Figure 8:** FedCMA latent space decision boundaries

We first evaluate Fed-CMA on a synthetic dataset to illustrate the conditional alignment in the latent space. We then compare Fed-CMA against other FL algorithms in the model heterogeneity setting on various image datasets.

**Synthetic dataset** We solve a 4-class classification problem, where we generate a synthetic dataset in $\mathbb{R}^2$ from a mixture of four Gaussian distributions and distribute them in a non-i.i.d. way among 4 clients (see Fig 4, 7). We use Linear(2,2) $\rightarrow$ Linear(2,2) $\rightarrow$ Linear(2,4) at every client for simplicity. After training, we plot the decision boundaries in the original and latent spaces (both $\mathbb{R}^2$). We see that the distributions and decision boundaries in the latent space are aligned for Fed-CMA (see Fig 8) resulting in much better decision boundary in the original space, even for workers with scarce data (see Fig 7). We empirically approximate $d_{\mathcal{V}\Delta\mathcal{V}}$ by training a classifier to discriminate between the latent space data of the four clients (motivated from Ben-David et al. (2010), see supplementary material for more details). As we observe from Fig 9a, as the training progresses the accuracy of the discriminator reaches that of a random classifier which means the latent space data among the workers are more aligned and it gets difficult to tell from which client the data came from.

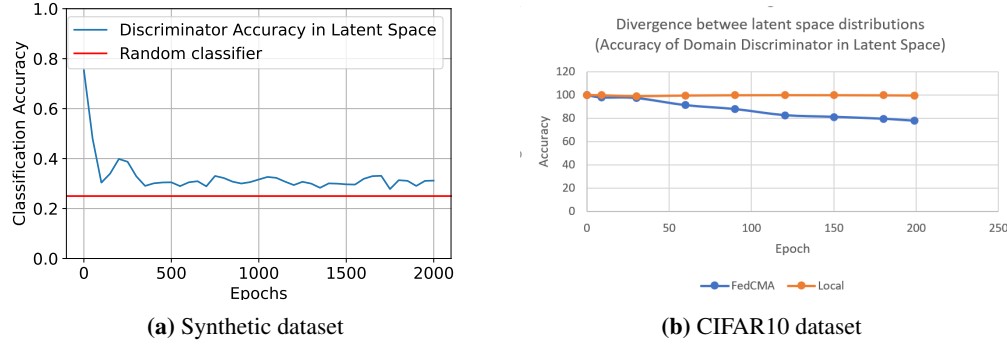

| (a) Synthetic dataset | (b) CIFAR10 dataset |

**Figure 9:** Discriminator classification accuracy (smaller the better)

**MNIST datasets**    We perform evaluation on MNIST, EMNIST and FEMNIST Caldas et al. (2018) datasets. For MNIST and EMNIST, we consider $M = 20$ clients and to simulate **data heterogeneity**, we follow Hsu et al. (2019) where a Dirichlet distribution $\text{Dir}_M(\alpha)$ is used to partition among the $M$ clients. For smaller values of $\alpha$, the data is more heterogeneous and for larger values of $\alpha$ the data is more homogeneous across the clients. For the MNIST and EMNIST datasets, we consider full client availability. In the FEMNIST experiment, we consider around 25% of the FEMNIST dataset and contains 923 clients with 50% client participation at each round.

Note that it is hard to compare the performance of Fed-CMA with earlier works such as Fed-ET Cho et al. (2022), FedDF Lin et al. (2020) and KT-pFL Zhang et al. (2021) since they utilize additional information in the form of a public dataset that is used for knowledge distillation. Moreover, these works may involve sharing the parameters with the server. Therefore, we do not provide a comparison with these works. While the main results of FedGen Zhu et al. (2021) require sharing all the model weights with the server, FedGen also accommodates limited parameter sharing (such as sharing only the classification layer weights $\mathbf{v}_i$). Therefore, in this section, we compare Fed-CMA with FedGen under the limited parameter setting. Additionally, we also provide a comparison against local training and against FedAvg which simply shares the final layer weights (called FedAvgSim in Pillutla et al. (2022)). We use the same network and setup used in Zhu et al. (2021) using the official implementation[1] and allowed multiple local client updates between communication rounds.

From the results in Table 2, we observe that Fed-CMA outperforms all the baselines in the limited parameter setting. The improvement offered by Fed-CMA over the other baselines is much higher in the high data heterogeneity (small $\alpha$) setting and the improvements diminish as $\alpha$ increases. This shows that in more practical settings, Fed-CMA can provide much needed gains to justify participating in the FL setup. We provide detailed experimental setup and ablations (marginal alignment vs conditional alignment, advantage of sharing mean embeddings) in the supplementary material.

| dataset | $\alpha$ | Local | FedAvg | FedGen | FedCMA |
|---------|------|-------|--------|--------|--------|
| MNIST | 0.02 | 76.28 (0.39) | 76.14 (0.48) | 75.95 (0.06) | **79.22** (0.74) |
| | 0.03 | 59.89 (0.26) | 59.82 (0.23) | 60.08 (0.61) | **64.03** (0.26) |
| | 0.05 | 61.36 (0.71) | 61.26 (0.70) | 60.82 (0.50) | **62.57** (0.39) |
| | 0.10 | 62.27 (0.48) | 62.26 (0.35) | 62.84 (0.07) | **63.94** (0.42) |
| EMNIST | 0.02 | 50.2 (0.10) | 49.77 (0.11) | 49.93 (0.06) | **51.0** (0.29) |
| | 0.03 | 47.95 (0.33) | 47.46 (0.24) | 47.5 (0.32) | **49.26** (0.15) |
| | 0.05 | 52.73 (0.35) | 52.2 (0.36) | 52.48 (0.33) | **53.96** (0.48) |
| | 0.10 | **47.31** (0.35) | 46.84 (0.29) | 46.35 (0.34) | 46.5 (0.32) |
| FEMNIST | - | 62.49 (0.64) | 63.50 (0.79) | - | **64.50** (1.2) |

**Table 2:** Comparison on MNIST, EMNIST and FEMNIST datasets with data heterogeneity and limited parameter sharing

---

[1]https://github.com/zhuangdizhu/FedGen

| Workers | pFedHN | pFedHN-PC | FedCMA |
|---|---|---|---|
| 10 | 88.34 | 89.05 | **91.36** |
| 50 | 83.62 | 83.62 | **84.24** |
| 100 | **82.73** | 80.81 | 80.92 |

**Table 3:** Comparison on CIFAR10 with data heterogeneity and limited parameter sharing

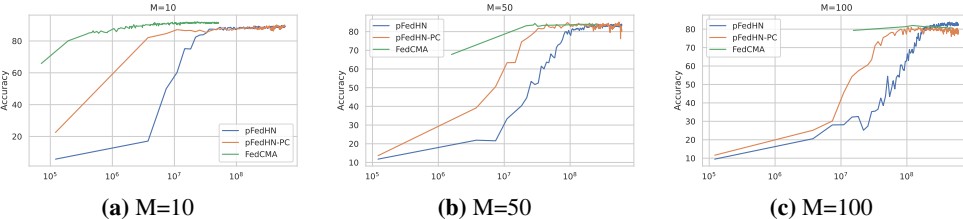

**(a)** M=10        **(b)** M=50        **(c)** M=100

**Figure 10:** Comparison of communication performed during training for CIFAR10 (x-axis) vs accuracy (y-axis)

**CIFAR-10 dataset** We also compare our work against the personalized FL algorithms pFedHN, pFedHN-PC in Shamsian et al. (2021) for the CIFAR-10 dataset. We use the same non-i.i.d. data split, networks and setup used in the official implementation of Shamsian et al. (2021)[2]. The comparison is provided in Table 3 and we observe that Fed-CMA outperforms the compared algorithms in two out of the three cases. Additionally, we also provide a comparison of the communication cost (in terms of number of parameters) for the three algorithms in Fig 10. We observe that Fed-CMA achieves similar accuracy as the other two algorithms despite requiring an order of communication lesser. Note that, the communication complexity of Fed-CMA at each round is $\mathcal{O}(K \times M \times d_e)$ irrespective of the dimension of the feature extractor weights. We also empirically approximate $d_{\mathcal{V}\Delta\mathcal{V}}$ by training a domain discriminator in the latent space and conclude from Fig 9b that the reduced accuracy of the domain discriminator is due to a better alignment in the latent space for Fed-CMA (which improves the generalization performance). Observe that for local training, the domain discriminator has 100% accuracy, implying that there is zero alignment of latent space distributions.

## 7 Conclusion and Future Work

In this work, we focused on the model heterogeneous FL setup without assuming a public dataset, without imposing restrictions on the choice of the model architectures and keeping the model architectures private. We proposed Fed-CMA based on conditional distribution alignment in the latent space in a federated way. We prove the convergence and generalization properties of the algorithm with emphasis on the role of the learned representations. These insights are helpful in designing better FL algorithms and the proposed algorithm can be used along with existing FL algorithms in the weight sharing setup, and/or use an adversarial loss to improve the performance.

Our current convergence analysis covers the case where each client performs only one local update and all clients participate in the training at every step. The impact of local communication steps on the convergence rate can be considered as future work, since this issue has deviated from the main topic (i.e., improved generalization) of this paper. Moreover, our work opens up several intriguing questions: (i) What level of model heterogeneity can be tolerated to achieve a reasonable improvement in the FL setup?, (ii) What is the role of the latent space in tolerating this model heterogeneity?

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
