# OpenReview forum: "Conditional Moment Alignment for Improved Generalization in Federated Learning"
_NeurIPS.cc/2022/Workshop/Federated_Learning — FL-NeurIPS 2022 Oral_

### Official Review · Reviewer_nfPq · 2022-10-11
**Good paper with solid theory and experiments**

This paper looks at FL from multiple heterogeneous clients who cannot share data due to confidentiality concerns and who might have data from slightly different distributions. The paper then proposes a nice solution that is theoretically well justified, and experimentally validates it.

Overall I thought this was a nice paper with both really good theory as well as experiments. I think it would be of interest to the workshop attendees.

---

### Official Review · Reviewer_zLcR · 2022-10-16
**The authors derived convergence results for addressing model heterogeneity via exploiting theoretical results in DA Peng et al. (2019).  Some theoretical analysis for their proposed Fed-CMA could be further investigated, and more typical heterogenous settings should be evaluated in the experiments.**

The authors derived convergence results for addressing model heterogeneity via exploiting theoretical results in DA Peng et al. (2019).  The authors showed the impact of model heterogeneity in the convergence bound in terms of the role of representations {uj}. Then, they propose Fed-CMA to minimize the divergence.

Comments:
1. Some of the insights are trivial, e.g., the generalization error decays with the participating client number.
2. The proposed Fed-CMA could be further explained, e.g., why first align the order moments of the class conditional mean, and how the proposed Fed-CMA improves the convergence rate. Some theoretical analysis in supporting their algorithm design could be investigated.
3. The Dirichlet distribution may not well represented some typical heterogeneity settings. More standard, such as non-IID (2) for cifar 10 and mnist datasets should be conducted to effectively evaluate the proposed Fed-CMA.

---

### Official Review · Reviewer_hz54 · 2022-10-17
**Principled idea for model heterogeneous FL with thorough experimentation**

In this paper the authors propose a technique that aligns classification weights and latent spaces of heterogeneous models to improve performance. The authors provide theoretical justification and insights along with thorough experimentation.

The text overall could be a little bit clearer as it packs a lot of information in sequence without first giving high-level directions to the arguments. The main table of results (Table 2) has no form of confidence interval, so it is hard to judge the significance of the (accuracy?) values. The metric was also not clear form the text.  Does +1, 2 or 3% in accuracy justify the use of this method? If there are other advantages, then they should make the main experiment section of the paper (and not the appendices). For instance, FedAvg drops about 18% when alpha goes from 0.02 to 0.10; FedCMA drops about 19% - so it has a higher (or about the same) relative drop on performance when going from less to more heterogeneous. How does this relate to the argument that it is actually more robust to heterogeneity?

Overall, I believe the method is a fair idea and the community would be better discussing its advantages and pitfalls. However, although possessing valuable theoretical and empirical insights, the paper could have been more focused and clearer in its presentation.

---

### Decision · Program_Chairs · 2022-10-20

Accept (Oral)